# Nonlinear Regularization Decoding Method for Speech Recognition

**DOI:** 10.3390/s24123846

**Published:** 2024-06-14

**Authors:** Jiang Zhang, Liejun Wang, Yinfeng Yu, Miaomiao Xu

**Affiliations:** College of Computer Science and Technology, Xinjiang University, Urumqi 830017, China; zhangjiang@stu.xju.edu.cn (J.Z.); yuyinfeng@xju.edu.cn (Y.Y.); xmm@stu.xju.edu.cn (M.X.)

**Keywords:** nonlinear Transformer, regularization attention, speech recognition, hybrid Transformer decoder

## Abstract

Existing end-to-end speech recognition methods typically employ hybrid decoders based on CTC and Transformer. However, the issue of error accumulation in these hybrid decoders hinders further improvements in accuracy. Additionally, most existing models are built upon Transformer architecture, which tends to be complex and unfriendly to small datasets. Hence, we propose a Nonlinear Regularization Decoding Method for Speech Recognition. Firstly, we introduce the nonlinear Transformer decoder, breaking away from traditional left-to-right or right-to-left decoding orders and enabling associations between any characters, mitigating the limitations of Transformer architectures on small datasets. Secondly, we propose a novel regularization attention module to optimize the attention score matrix, reducing the impact of early errors on later outputs. Finally, we introduce the tiny model to address the challenge of overly large model parameters. The experimental results indicate that our model demonstrates good performance. Compared to the baseline, our model achieves recognition improvements of 0.12%, 0.54%, 0.51%, and 1.2% on the Aishell1, Primewords, Free ST Chinese Corpus, and Common Voice 16.1 datasets of Uyghur, respectively.

## 1. Introduction

As one of the most convenient and direct means of communication between individuals, voice plays a crucial role in today’s society. Speech recognition technology has been widely applied in various fields, encompassing scenarios such as smart homes and text input. It enhances efficiency in both personal and professional spheres and provides people with a more convenient and intelligent experience. Current speech recognition research is divided into audio-based and audiovisual speech recognition. Audiovisual recognition [1,2,3] is primarily conducted with a limited vocabulary and requires video speech data. Existing audio speech recognition technology primarily targets languages with abundant training resources, leaving limited research on low-resource languages like Uyghur. Uyghur and Chinese are widely used in the Xinjiang region of China and are employed in various public settings, entertainment videos, and news and television dramas. Compared to languages with abundant training resources, Uyghur has relatively limited speech data, posing challenges to existing large-scale parameter models.

In traditional speech recognition models [4,5,6], the training of each module, such as the acoustic and language models, is conducted independently. This independence complicates the optimization process, resulting in increased training complexity. Moreover, it may require prior knowledge, which is disadvantageous for advancing automatic speech recognition. With the development of deep learning, end-to-end speech recognition has gradually become mainstream in the field. End-to-end speech recognition directly maps input speech signals to corresponding text transcriptions or text representations, eliminating intermediate processing stages and reducing the need for prior knowledge. In end-to-end speech recognition, models are typically divided into two components: an encoder and a decoder. In the encoder, the Conformer architecture [7], which combines global and local information, has emerged as a new baseline. Decoders typically include Connectionist Temporal Classification (CTC) [8], attention decoding [9], and hybrid decoders [10,11], each with its advantages. CTC decoding offers faster speed but overlooks contextual relationships, whereas attention decoding has a slower speed but higher accuracy. Consequently, hybrid decoder [12] has gradually become the optimal solution.

The representative method in the hybrid decoder is the attention-rescoring decoding approach. During the training process, the losses of both decoding methods are fused to jointly guide the optimization of the encoder. In the inference process, the CTC beam search decoding results are fed back into the attention mechanism for rescoring, aiming to achieve better results. However, attention decoding follows an autoregressive decoding manner, decoding sequences linearly from left to right. Despite bidirectional [13,14] Transformers providing reverse decoding information, the decoding sequence remains relatively monotonic, struggling to extract semantic information effectively, especially on small datasets where training resources are relatively scarce and cannot meet the demands of current popular model training. Meanwhile, attention models are not well-suited for the rescoring approach because, during beam search, many decoding errors may occur. Sending these error-prone data to the attention model could lead to error accumulation. In other words, early errors in beam search output may significantly impact subsequent attention outputs, resulting in cumulative errors. Moreover, existing end-to-end models typically have massive parameter counts, requiring substantial computational resources during training and posing challenges for lightweight deployment [15].

We propose the Nonlinear Regularization Decoding (NLRD) Method for Speech Recognition to address the challenges above. Addressing the suboptimal performance of the Transformer on small datasets and the limitations of left-to-right decoding, we propose a nonlinear Transformer (NLT) module. This module breaks the traditional decoding sequence constraints, allowing for multiple decoding of different character sequences. As a result, correlations can be established between any characters, better guiding the encoder to extract effective features, thereby achieving more comprehensive model training. We introduce a regularization attention (R-attention) module to address the error accumulation challenge during attention-rescoring decoding. By regularizing the attention score matrix, this module effectively controls the impact of errors in early beam search decoding results on subsequent attention outputs, successfully avoiding the challenge of error accumulation. To address the high complexity of the NLRD model, we propose a model named the NLRD-tiny. The architecture only utilizes the NLT module and a single-layer R-attention module while reducing the weight of the CTC branch in the attention-CTC dual-branch decoder. This focus on semantic information helps mitigate potential information loss caused by the independence assumption of CTC, thereby improving the effectiveness of CTC decoding.

The main contributions of this paper are as follows:This paper introduces an NLT module to improve Transformer performance on small datasets and overcome left-to-right decoding limitations.This paper proposes an R-attention module to mitigate error accumulation in attention-rescoring decoding.Proposing the NLRD-tiny model, this paper simplifies the NLRD model’s high complexity by using only the NLT module and a single-layer R-attention module.

## 2. Related Work

End-to-end speech recognition has emerged as a prominent technology in speech recognition, directly mapping speech signals to text transcriptions and eliminating the cumbersome intermediate processing steps of traditional models. With the continuous advancement of deep learning techniques, the hybrid decoder based on Transformer has become one of the most researched topics in recent years. This decoder combines the powerful representation learning capabilities and fusion mechanisms of a Transformer, bringing breakthroughs to speech recognition tasks. Next, we will introduce these two important topics separately.

### 2.1. End-to-End Speech Recognition

End-to-end speech recognition uses a single model that directly maps input acoustic features to a text representation. The end-to-end speech recognition model consists of the encoder and the decoder. Combining global and local information in the encoder stage is an important approach [16,17]. In the field of speech recognition, the mainstream model is the Conformer. The Conformer [7] model combined the advantages of convolutional neural networks (CNNs) and Transformer models, using CNNs to extract detailed features and Transformer for global modeling, significantly improving speech recognition accuracy. Conformer has become the new baseline system for speech recognition. Following that, Efficient Conformer [18] and UCONV-CONFORMER [19] improved the complexity of Conformer to enhance the model’s speed. Efficient Conformer effectively reduces the complexity of the Conformer model by introducing downsampling modules and reducing the dimensionality of the self-attention process. UCONV-CONFORMER reduces the input sequence length by 16 times and adds internal CTC loss to aid model convergence. These advancements have positively impacted the development of speech recognition technology.

The decoder is the core of end-to-end speech recognition. The CTC is a groundbreaking work [20]. In the CTC decoding process, developing the current decoding is selected at each step, known as greedy search. Under the greedy search algorithm, CTC decoding is very fast, but CTC assumes that each frame is independent. In reality, there is a wealth of semantic information between speech frames. Speech recognition models based on long short-term memory networks (LSTMs) decoding have been proposed to address this issue. LSTMs have gained prominence in speech recognition tasks, with models like DeepSpeech [21] and DeepSpeech2 [22] adopting LSTM-based architectures. Nevertheless, LSTMs only capture forward information. To enhance the accuracy of speech recognition, Jorge introduced Bidirectional LSTMs. Bidirectional LSTMs can capture information from both forward and backward directions, thereby improving the precision of decoding [23].

The Transformer decoder is commonly used in end-to-end speech recognition models. It typically comprises two types of attention: self-attention modules and cross-attention modules [24]. The self-attention module is employed to encode the output from the previous step. In contrast, the cross-attention module captures the relationship between the previous step’s output and the encoder’s output, enhancing the correspondence between sequences. The Speech Transformer [9] not only focuses on temporal information but also emphasizes frequency information, extracting more effective features. Additionally, it adopts a more lightweight model, accelerating training speed and holding significant importance for advancing speech recognition. Subsequently, Transformer decoders have undergone various improvements, such as refining positional information [25], incorporating positional information into encoder outputs [26], and enhancing bidirectional semantic information [13,14,27]. However, regardless of the type of decoder, there are certain limitations.

Due to the limitations of a single model, the field of speech recognition is gradually adopting hybrid decoding models [10,11,28]. Our model differs from other end-to-end models in that it incorporates a nonlinear Transformer, enabling it to capture semantic information more comprehensively while further optimizing the extraction capability of effective features.

### 2.2. Hybrid Decoder Based on Transformer

Fusion models can often achieve better results [29]. This is because fusion enables the incorporation of more information, forming a more powerful feature representation. Fusion can involve combining different modalities features, merging features from multiple networks [30], and so on.

The decoder also employs fusion in speech recognition by sharing the encoder’s input among different decoders. After calculating the loss, the gradients are propagated backward to update the encoder. This approach offers advantages by mitigating the inherent shortcomings of a specific decoder, facilitating the model in learning more useful features. The hybrid decoder includes methods such as CTC + RNN [31], CTC + Transformer [11,32], and others. Due to the potential issue of gradient vanishing with LSTMs, much research favors hybrid decoder models using CTC + Transformer. Commonly used decoders in speech recognition include the Transformer and CTC. The Transformer is a stacked attention model that maximizes the acquisition of global information. Additionally, as an autoregressive model, the Transformer naturally excels in sequence tasks like speech. However, hybrid decoders combine CTC and Transformer losses to maximize the model’s generalization performance [33]. As a result, hybrid decoders have become increasingly popular.

To implement the aforementioned recognition models and replicate these advanced experimental results, researchers in the field of speech recognition have released several open-source toolkits, including Kaldi [34], Fariseq [35], ESPnet [36], and, more recently, Wenet [10,28]. The Wenet toolkit adopts an advanced decoding approach known as hybrid decoders. In the hybrid decoders, the Transformer and CTC share the same encoder, and the losses of both decoders are jointly optimized. During the inference process, Wenet is distinctive in employing a method called attention-rescoring. In this method, the top N recognition results from beam search are fed back into the attention mechanism for rescoring, thereby achieving superior decoding results. The difference between our hybrid decoder and others lies in the restriction imposed on attention in our decoder. This effectively mitigates the impact of CTC’s error outputs on the attention model within the hybrid decoder.

## 3. Methodology

The NLRD model we proposed adopts an encoder–decoder architecture. The encoder utilizes the Conformer model. The decoder consists of CTC decoding and Transformer decoding. In contrast, the Transformer decoding part of the decoder employs our designed NLRD, which consists mainly of NLT and R-attention modules. Additionally, we introduce a streamlined variant, NLRD-Tiny, to facilitate lightweight processing. The overall workflow of our model is illustrated in Figure 1.

### 3.1. The NLT Module

In traditional Transformer decoding models, the decoding process is executed sequentially. That is, starting from a given initial token, multiple characters are identified one by one in sequence. The decoder determines the next decoding output based on the output of the previous step (or the initial token) and the output of the encoder. However, by breaking the constraints of sequential decoding, the model can better learn the relationships between arbitrary characters and extract more semantic information. In other words, this model can decode arbitrary sequences, which means multiple effective decodings can be performed on the same data, i.e., multiple effective training sessions can be conducted on the same data. This partially overcomes the limitation of dataset size. Compared to sequential decoding methods, this approach can better train small-scale datasets. The input of any sequence is implemented through different masks. By employing various masks, theoretically, we achieve input for any sequence. However, in reality, the input sequence itself remains unchanged. It is important to note that our decoding process is independent, meaning each mask corresponds to a decoder, and the decoding sequence is disrupted. However, the decoding process still relies on the previous output to determine the current output without accessing future decoding information during the current decoding process. This aspect is consistent with the concept of bidirectional Transformer [14].

The difference from the bidirectional Transformer [13,14] lies in the unrestricted nature of the NLT decoding process in this paper, which can decode in any order without directional constraints. Specifically, while bidirectional LSTMs use both forward and backward information simultaneously during decoding, this paper allows decoding to proceed independently in any direction. This includes forward decoding, the conventional decoder, backward decoding, decoding from back to front, and decoding in other specific directions. Subsequently, we compute the average loss of the outputs from these nonlinear decoders. This enables the extraction of deeper semantic information through decoders in different directions and guides the encoder to perform better feature extraction through backpropagation. As a result, we can comprehensively train the Transformer model, enabling it to achieve excellent performance even on small datasets. The specific computational process is illustrated in Figure 2.

We have designed a masking method to enable the Transformer decoder to decode in any direction. Assuming the output sequence is $Y=\{y_{1},y_{2},...,y_{n}\}$, there can be *K* different permutations, where K=n!. To save training time, we usually set *K* as a hyperparameter, and in our experiment, we chose K=6. Next, we illustrate this with a simple example for ease of demonstration, setting n=6 and K=3. Therefore, *Y* has 6 permutations, and we randomly select 3 of them. One is fixed as forward decoding, i.e., $\left\{y_{1},y_{2},y_{3}\right\}$, to ensure the correct linguistic order. Then, we randomly choose the other two, for example, $\left\{y_{1},y_{3},y_{2}\right\}$ and $\left\{y_{2},y_{1},y_{3}\right\}$. When K=3, the loss of Transformer decoding can be expressed using Equation (Equation 1). L_attention represents the loss of the Transformer, which is subsequently cumulatively weighted with the CTC loss to form the final loss.
(1)$$\begin{array}{c} \mathrm{L\_attention}=\mathrm{loss}(y_{1,2,3}^{\prime },y_{1,2,3})\\ +\mathrm{loss}(y_{2,1,3}^{\prime },y_{2,1,3})\\ +\mathrm{loss}(y_{2,3,1}^{\prime },y_{2,3,1}) \end{array}$$

Our practical experiments perform a reversal operation after obtaining *K* decoding sequences to strengthen the relationship between the preceding and succeeding sequences. Specifically, if there are *K* decoding sequences, we reverse these *K* sequences and then reintroduce these reversed sequences into the decoding sequences. To illustrate with a specific example, if *K* equals 6, we effectively perform 12 decoding operations. After these 12 decoding operations, we calculate the average of the obtained losses, which is then weighted and summed with the CTC loss. This process helps optimize the model, enabling it to understand better and capture the correlation between preceding and succeeding sequences.

### 3.2. The R-Attention Module

We know that Wenet’s attention-rescoring decoder takes the decoding results of beam search and passes them through the attention model for rescoring. Each output has ten candidate outputs when selecting the top 10 decoding results of the beam search. There is a significant gap between these outputs and the actual output, which can substantially impact the decoding of the attention model. Let us illustrate this with an example: Suppose there is a speech input with the label “I love deep learning”, and selecting the top 2 results from beam search might be “I love deep learning” and “I dove deer learning”. In the process of the attention-rescoring decoder, due to the autoregressive nature, the incorrect input “dove” will have a persistent negative impact on subsequent outputs, leading to even larger errors. To address this challenge, we adopt a specific regularization method by controlling the attention scores within a certain range and emphasizing the current input’s impact on the results. The main difference between the R-attention and regular attention models lies in the regularization of attention scores. We provide a detailed description through Equations (2) and (3).
(2)$$\begin{array}{cc} D=score-\mathrm{diagonal}(score) & \; \end{array}$$
(3)$$\begin{array}{cc} \; & \mathrm{R\_score}=\mathrm{sigmoid}(D)\times (1-\mathrm{sigmoid}(D)) \end{array}$$

We calculate the difference between each row of the original attention score matrix and its corresponding diagonal data using Equation (Equation 2). Then, we apply Equation (3) to activate the obtained difference matrix in reverse. This operation aims to subtract the diagonal values of attention scores from their corresponding rows, resulting in an attention matrix with zeros along the diagonal. This matrix has zero values on the most crucial self-correlations. Through reverse activation, we can focus the attention scores on key information. Additionally, the closer the position is to the current decoding position, the stronger the correlation, while positions farther away experience suppression of correlation. In this way, we successfully reduce the accumulation of errors caused by incorrect beam search decoding results. During the training, the input is the output of the previous step, so it is not limited by the calculation of relevance. However, during the inference, when using the attention-rescoring decoding method, the input is the result of CTC decoding, which may contain a large amount of erroneous information. Our R-Attention will restrict this information to prevent errors from accumulating in the subsequent output.

### 3.3. The NLRD-Tiny Model

In the training process of our NLRD-tiny model, we employed a joint loss, which includes frame-level CTC decoding loss and R-attention decoding loss. Unlike the attention-rescoring method, our attention mechanism only utilizes a single layer of R-attention. As the NLT module in our decoder is already capable of extracting sufficient contextual information, in the NLRD-tiny model, we reduce the decoder to a single layer. This effectively reduces the model’s parameter count, providing insights into the model’s lightweight and further validating the effectiveness of our NLRD model. The loss calculation formula we used is given by Equation (Equation 4) during the training process.
(4)$$\begin{array}{cc} \mathrm{Joint\_Loss}(x,y) & =CTC\_weight\times L\_\mathrm{CTC}(x,y)\\ \; & +(1-CTC\_weight)\times L\_\mathrm{attention}(x,y) \end{array}$$
*x* represents the output of the shared encoder, and *y* represents the labels. Combining CTC and attention loss can result in superior performance and generalization compared to using either individually. The combined loss effectively provides complementary information for two decoding approaches. During the inference process, we exclusively use the CTC decoding method and employ a greedy search decoding approach, effectively reducing the model’s parameters and improving inference speed.

The advantage of the NLRD-tiny model lies in reducing the Transformer decoder from the original Nonlinear Regularization decoding method to a single layer. Our R-attention module decodes multiple times during training, thereby learning more semantic information. This helps to compensate for potential information loss due to the CTC independence assumption, thereby improving the effectiveness of decoding. It is worth noting that we only use forward decoding during the inference process, similar to regular attention models, and it does not increase the inference time. The model structure of the NLRD-tiny model is illustrated in Figure 3.

In Figure 3, solid lines represent the training process, while dashed lines depict the inference process for attention-rescoring. The shared encoder is a shared encoder utilized for CTC and attention decoding. It consists of a 12-layer conformer encoding module.

## 4. Experiment and Analysis

### 4.1. Experimental Setup

We utilized three Chinese datasets, namely Aishell1 [37], Prime (Primewords Chinese Corpus Set 1) [38], and ST (Free ST Chinese Mandarin Corpus). Additionally, we employed a Ug (Uyghur language) dataset from Common Voice 16.1 for Uyghur speech recognition. These four datasets each consist of just over 100 h of data. Compared to larger datasets like LibriSpeech, which contain over 1000 h of data, these four datasets are considered small. Aishell1 is a Chinese speech recognition dataset provided by iFLYTEK under the Chinese Academy of Sciences’ Institute of Automation. The dataset includes training, validation, and test sets, making it a high-quality collection for speech recognition. It involves recordings from over 100 individuals, totaling 170 h. Primewords Chinese Corpus Set 1, released by Shanghai PuLi Information Technology Company Limited, contains 178 h of data. The ST dataset was recorded in a quiet indoor environment. It consists of 855 speakers, with each speaker having 120 utterances. Humans have transcribed and meticulously checked all utterances to ensure transcription accuracy. Common Voice is an open project initiated and maintained by Mozilla, comprising a multilingual speech dataset. Among them, the Uyghur speech recognition dataset in Common Voice 16.1 totals 180 h.

Unlike Aishell1 and Common Voice, Prime and ST are not explicitly divided into training, validation, and test sets. Following the proportions of the Aishell1 dataset, we randomly split both datasets between a ratio of 85% for training, 10% for validation, and 5% for testing. In the Ug dataset, there are 147 h of validated data. Excluding the test and validation datasets, we incorporate all remaining data into the training dataset.

The generation of vocabulary varies among the four datasets. As Chinese is based on individual characters, using single Chinese characters effectively reduces the vocabulary size. Therefore, for Chinese datasets, we use single Chinese characters as the tokenization standard and include “black” and start/end tokens in the vocabulary. Ultimately, the vocabularies for the Aishell1, Prime, and ST datasets contain 4233, 5800, and 4187 items, respectively. We employ the publicly available Byte Pair Encoding (BPE) method in the UG dataset for vocabulary generation. BPE [39] is a common algorithm for subword tokenization in natural language processing. By progressively merging frequent character sequences into larger subword units, BPE effectively captures morphological information and statistical patterns in the training data. It also helps handle rare or unseen words by breaking them into smaller subword units. In our experiments, we divide the training data into 5000 subword units. The differences in vocabulary among the four datasets result in slightly different model sizes, albeit with minor variances.

The common evaluation standards for speech recognition are word error rate (WER) and character error rate (CER), where lower CER and WER values indicate better performance, as they reflect lower error rates. In Chinese speech recognition, the segmentation system significantly influences the recognition results of WER. Therefore, CER is commonly used as the evaluation standard in Chinese speech recognition. We adopt CER as the evaluation metric for Chinese speech recognition, while for Uyghur language, we utilize WER as the evaluation criterion. The computation formulas for CER and WER are represented by Equation (Equation 5).
(5)$$CER=\frac{S+D+I}{N}$$
*S* represents substitution errors, *D* represents deletion errors, *I* represents insertion errors, and *N* denotes the total number of words in the reference text. We will list and evaluate *S*, *D*, and *I* separately for easy comparison.

In our experiment, we chose a server CPU equipped with AMD Ryzen 5 5600 (Advanced Micro Devices, Inc. (AMD), Santa Clara, CA, USA), and configured a graphics card with 24 GB of graphics card memory, namely the NVIDIA GeForce RTX 3090 Ti (NVIDIA, Santa Clara, CA, USA). In our experiment, the allowable batch size varies due to the difference in speech lengths between the four datasets. The batch size for the PrimeWords dataset is set to 24, while for other datasets, it is set to 36. Table 1 shows the settings for other hyperparameters. Num_Conformer refers to the number of layers in the encoder. Since our model utilizes the Conformer as the encoder, it is directly represented as Num_Conformer. Num Decoder indicates the number of layers in the decoder. Epoch refers to the training batches, and LR denotes the learning rate. Due to the complexity of Chinese and Uyghur languages and differences in feature distributions, the learning rate requirements vary during the training process for the two languages. Therefore, the table lists separate hyperparameters for the two language datasets. Dff refers to the dimension of the hidden layer in the feedforward neural network, Num_Mel refers to the input feature dimension, and Accum_Grad refers to accumulated gradient. Frame_Length and Frame_Shift are the settings for speech framing, which are common in speech recognition models. They are typically set to 25 ms and 10 ms, respectively. Additionally, the Spec_Aug toolkit (https://github.com/pyyush/SpecAugment, 1 May 2024) is utilized for data augmentation in speech recognition. Spec_Aug is an open-source toolkit for data augmentation in speech recognition tasks. Gradient accumulation is set to 4.

### 4.2. Comparative Experiments

Our paper compares 11 speech recognition methods. The Speech_Transformer and Transformer models both use Transformer-based encoders and decoders. RNNT and BAT use RNN-based decoders, while NST is a semi-supervised model. Whisper, developed by OpenAI, is a large multi-language and multitask model based on the Transformer architecture, with Whisper (large-v3) being its improved version. Paraformer and Efficient Conformer are based on the Conformer model, with Paraformer (U2) being an improved version using the U2 [10] architecture. The Conformer_Transformer is the baseline model, employing a Conformer encoder and a Transformer decoder. The experimental results are shown in Table 2.

The experimental data in Table 2 demonstrate that our method, NLRD, achieved the best performance compared to other methods. This success is attributed to our designed NLT and R-attention modules. The NLT module decodes character sequences multiple times to establish correlations between characters, thus guiding the encoder to extract more effective features. The R-attention module effectively controls the impact of errors in early beam search decoding results on subsequent attention outputs, preventing error accumulation. Compared to the baseline methods, our NLRD model reduced error rates on the Aishell1, Prime, ST, and Ug datasets by 0.12%, 0.54%, 0.51%, and 1.2%, respectively. The improvement on the Aishell1 dataset is relatively modest because the baseline already performs well on this dataset, leaving limited room for improvement. In contrast, the significant improvement observed in the Ug dataset is due to the frequent occurrence of affix errors in Ug language recognition. Our model effectively extracts contextual information, thereby avoiding these errors.

Table 2 shows that the Efficient Conformer also performs excellently. It combines CNN and Transformer in the encoder, providing better detail extraction capability. Thus, it stands out, particularly on relatively small datasets. Next, RNNT also performs well. It enhances CTC by modeling textual information, with outputs relying on acoustic and historical information. RNNT integrates decoding and encoding information during decoding, retaining richer features for more accurate recognition results, though it incurs significant training costs. BAT is a lightweight improvement on RNNT but requires substantial training data. Therefore, its performance on smaller datasets is inferior to that of RNNT. Paraformer, a non-autoregressive model, enhances contextual modeling via the GLM [44] sampler but underperforms compared to autoregressive models in this experiment. Paraformer (U2) also employs non-autoregressive decoding and adds CTC loss for joint optimization, improving the encoder’s feature extraction capability. Although better than Paraformer, it still lacks contextual modeling capability. NST, a semi-supervised model, shows moderate performance as no additional unlabeled data was used for pre-training in the experiment. The Speech Transformer, merely applying Transformer to speech recognition without further optimization, performs poorly. The Transformer in Wenet employs the U2 architecture, combining CTC and Transformer losses, optimizing the traditional Transformer architecture, and thus outperforming the Speech Transformer. Whisper (base) is a large model known for its multilingual and multitasking capabilities but falls short in single-task performance compared to conventional models. Whisper (large-v3) includes more parameters and training data, performing better on Aishell1, Prime, and ST than Whisper (base). However, as it is designed for multilingual and multitasking purposes, its single-task performance is still not outstanding and is unsuitable for low-resource languages like Uyghur.

To demonstrate the capability of our model in extracting contextual information and implementing lightweight models, we have introduced a tiny model. Compared to the NLRD model, the NLRD-tiny model has only one layer in its decoder. In Table 2, it can be seen that the performance of our NLRD-tiny model remains superior to the baseline system. Compared to the baseline, our tiny model reduced error rates of 0.06%, 0.29%, 0.31%, and 0.58%, respectively. This is because our model, during the training process, utilizes forward and backward information and incorporates various other random sequence information and their reverse counterparts. This comprehensive extraction of semantic information enables our model, in contrast to the baseline’s six-layer decoder, to have a more nuanced handling of sequence relationships.

### 4.3. Ablation Study

To validate the effectiveness of the proposed approach in this paper, we conducted ablation experiments on the NLT module and the R-Attention module individually. In our NLT module, the number of decoding steps, denoted as *K*, significantly impacts the experiments. We found that increasing the value of *K* means performing more decoding steps during the training process, resulting in a slower training speed but potentially achieving better decoding results. Conversely, when *K* is small, it approaches the traditional Transformer model. We conducted ablation experiments on different *K* values. At the same time, we also conducted an ablation study on the impact of R-attention on the model. The experimental results are presented in Table 3. K=6 indicates the use of the NLT module, with the *K* value within the module set to 6. ✔ represents using this module, while ✗ represents not using this module, and the rest of the table follows this rule. The results on the Aishell1 dataset are denoted as Aishell1, while those on the Primewords dataset are denoted as Prime. In Table 3, the following apply: Encoder represents encoder layers; Decoder represents decoder layers; attention-rescoring, CTC-Greedy, and CTC-Beam, respectively, refer to the CER using attention-rescoring, CTC greedy search, and CTC beam search decoding methods. The CTC Beam has a beam size of 10.

Table 3 shows that both modules play a positive role in the overall system. A comparison experiment between the first and second rows of Table 3 reveals that, when K=3, the performance is slightly inferior compared to K=6. This is because, with increased decoding steps, the system can better capture the correlation between texts, extracting more semantic information. Therefore, as the value of *K* increases, the effectiveness of speech recognition correspondingly improves. However, it is worth noting that as *K* increases, the number of decoding steps also increases. Excessive decoding steps can impact training speed, and beyond extracting sufficient semantic information, further increasing *K* does not necessarily enhance recognition performance. In our experiment, an error exceeding the memory capacity will occur if we set K=9. Contrasting the experiments in the first and fourth rows of Table 3 indicates that, when K=6, the model utilizing the R-attention module demonstrates superior performance, reducing the CER from 4.52 to 4.49 on the Aishell1 dataset and from 12.51 to 12.36 on the Primewords dataset. Certainly, the third and fourth lines also indicate that the effectiveness persists when the two modules are used separately.

We know that network depth has a significant impact on experimental results. Our model excels at extracting features in models with small datasets and low parameter counts. To further demonstrate the effectiveness of our approach, we reduced the baseline model’s 12-layer encoder to 6 layers and the 6-layer decoder to 3 layers, resulting in an overall reduction of approximately half of the model parameters. Despite this halved parameter count, our NLT and R-attention modules still exhibit significant positive effects. The experimental results are presented in Table 4. The other parameter specifications are the same as Table 3, except that the data in Table 4 are all based on the results of a model with 6 layers in the encoder and 3 layers in the decoder. Table 3 has already verified that the model performs best when *K* is set to 6. Here, we do not further investigate the impact of different values of *K* on the results; all *K* values for the NLT modules are set to 6.

From Table 4, we can observe that even with only half of the parameter count, our model still achieves a CER of 4.83 and 13.26, showing a 0.22 and 1.44 improvement compared to the 6-layer encoder and 3-layer decoder model without our module. Compared to large models, our small model shows a more significant improvement. This also validates that our optimization method has a more pronounced effect on small models. The variations in parameters presented in Table 3 and Table 4 are due to the different output sizes of the two datasets. The Aishell1 dataset has 4232 character outputs, while the Primewords dataset has 5800 characters.

After confirming the effectiveness of the two modules, we set *K* = 6 and proceeded to validate our NLRD-tiny model. The model combines our NLT module with an R-attention module and adjusts the decoder layer to a single layer. This reduces parameter count and allows the R-attention to compensate for the loss of contextual information during CTC decoding, thereby improving the effectiveness of the CTC decoding process. The experimental results are presented in Table 5.

Especially noteworthy is that, after reducing the layers of the R-attention decoder, our model achieved a CER of 4.72 and 13.07 in the CTC greedy search decoder, an improvement of 0.3 and 1.91 compared to the baseline. However, the attention-rescoring method only showed an improvement of 0.23 on Aishell1 and 1.62 on Primewords. This indicates that our NLRD-tiny model exhibits more noticeable effects during CTC decoding. This is because our NLT module successfully extracted semantic information, compensating for the information loss caused by the CTC conditional independence assumption. At the same time, we lowered the CTC loss weight, allowing the R-attention to play a more effective role. From the experimental results in Table 5, NLT’s impact on the model is more significant when using lightweight decoding. When only removing NLT, the error rate of the model’s attention-rescoring increased by 0.12 and 0.6, respectively. In contrast, when only R-attention is removed, the error rate of the model’s attention-rescoring only increases by 0.01 and 0.12, respectively. However, the error rate of the CTC beam search (10) increased by 0.07 and 0.45, respectively. When not using both NLT and R-attention simultaneously, the error rate of attention-rescoring increased by 0.23 and 1.62, respectively. This increase is more significant than the impact of removing any single module alone, indicating that using these two modules simultaneously can better perform feature extraction and error correction.

Our NLRD-tiny model demonstrated improvements of 0.23 and 1.62 in attention-rescoring while achieving enhancements of 0.30 and 1.91 in the CTC-Greedy method. These results indicate that our NLRD-tiny model has a more pronounced positive impact on CTC decoding outcomes. This is because a single layer of R-attention has a relatively minor impact on optimizing the decoder. However, the optimization effect on the encoder did not diminish due to multiple decoding iterations, thereby extracting more optimal shared encoder features and consequently improving the CTC decoding performance. This result is of significant importance for lightweight models. Our NLRD-tiny provided a direction for the development of lightweight models.

The parameters of the NLRD-tiny model are provided in Table 5, showing a parameter count of 39.5 million compared to the baseline system’s 47.4 million, representing a reduction of approximately 17% in parameters. In other words, this implies that our NLRD-tiny model, while reducing parameter count by 17%, has also improved recognition performance. This further demonstrates the finer sequence relationship extraction capability of our NLRD model.

### 4.4. Visualization

We visualized the loss plot during the training process, distribution plots of various recognition errors, and line charts illustrating the reduction in parameter count and CER for our NLRD-tiny model. For conciseness and representativeness, our visualizations also showcase results from different datasets.

We plotted the loss in Figure 4 for comparison to demonstrate our results. The figure illustrates the validation loss on the Aishell1 dataset. Since different models employ different loss calculation methods, we have not included the losses of other state-of-the-art models. Instead, we present our baseline, which is the loss of the Conformer–Transformer model, along with the loss of our DLRD model. The figure shows that our validation loss is significantly lower than that of the original baseline system, highlighting the effectiveness of our model. Figure 4 also indicates that the loss reduction in our model is more stable. The baseline model exhibits early training fluctuations, attributed to the warm-up training approach where the learning rate gradually increases at the beginning of training. However, our model does not show such fluctuations. This is because our model employs a nonlinear decoding method, utilizing average loss, effectively preventing the model from becoming stuck in local optima.

Additionally, Figure 5 illustrates the distribution of recognition errors. In Aishell1, comprising 104,762 characters, the figure depicts the distribution of various errors. For the CTC decoding method, our model’s improvements over the baseline primarily focus on reducing substitution and insertion errors, with decreases of 65 and 12 character errors, respectively. This is due to the NLT method we designed, which uses 12 different decoding sequences for the same data during training. This enhances the contextual relationships between characters, allowing our model to learn semantic information more accurately and reduce recognition errors caused by the lack of semantic information in the CTC method. Additionally, since deletion errors are relatively few in our model and the baseline method, the improvement is less noticeable but still present. For the AR decoding method, our improvements over the baseline are even more significant, with reductions of 100 and 20 character errors for substitution and insertion errors, respectively. This is because we designed an R-Attention mechanism in addition to the NLT method, which optimizes the AR decoding strategy. Regularizing the CTC decoding results mitigates the impact of CTC errors on AR decoding. Therefore, the improvements with the enhanced AR decoding strategy are more pronounced.

Our NLRD-tiny model reduced the parameter count and lowered the error rate in speech recognition. For a more convenient presentation, we also visualized the parameter count and CER of our NLRD-tiny model, as shown in Figure 6. The data in the figure is obtained from the Prime dataset, and the CER here is calculated using CTC’s greedy decoding. This is because we aim for our NLRD-tiny model to be sufficiently fast during inference. Therefore, our NLRD-tiny model tends to favor the use of CTC decoding during inference. As observed, while reducing the number of parameters, there is also a decrease in CER. This further demonstrates the effectiveness of our model, as we managed to reduce the parameter count from 47.4 to 39.5, resulting in a 17% reduction, and concurrently, the CER decreased from 13.49% to 13.07%.

## 5. Conclusions

In light of the current challenges in speech recognition, such as insufficient training data, error accumulation, and challenges in deploying existing large speech recognition models in lightweight scenarios, we propose an NLRD model. NLRD adopts an encoder–decoder architecture. The encoder utilizes the Conformer model, while the Transformer decoding part of the decoder employs our designed NLRD, consisting mainly of the NLT and R-attention modules.

The NLT module allows the model to decode in any direction, breaking the constraints of traditional decoding sequences and thus acquiring richer semantic information. This module also ensures that Transformer models receive sufficient training in situations with limited training data. The experimental study is based on Wenet, which employs a joint decoder but faces challenges with error accumulation. The severity of error accumulation lies in the fact that early prediction errors may lead to a chain reaction, causing subsequent predictions to deviate from the true values and ultimately resulting in cumulative errors over time. To address this challenge, the R-attention module is introduced, which uses a novel regularization method to mitigate the impact of early decoding errors on subsequent attention outputs during beam search. Finally, we explored the lightweight model NLRD-tiny. Even with the application of NLT and a single-layer R-attention module, the lightweight model still performs well. After reducing 17% of the parameters, its accuracy remains superior to the baseline model.

The method presented in this paper also has certain limitations. Specifically, the training process of the NLRD method involves multiple decoding steps, leading to a decrease in training speed and an increase in GPU memory consumption. The extent of this increase is also influenced by our hyperparameter *K*—a larger *K* implies more decoding steps, resulting in higher GPU memory requirements and slower training speeds. However, the inference process does not involve multiple decoding steps, so inference speed remains unaffected. To address this limitation, we can mitigate it by appropriately reducing the number of decoder layers. Our experiments provide corresponding data, demonstrating that even with a reduced number of decoder layers, our NLRD method can still achieve satisfactory results.

## Figures and Tables

**Figure 1 sensors-24-03846-f001:**
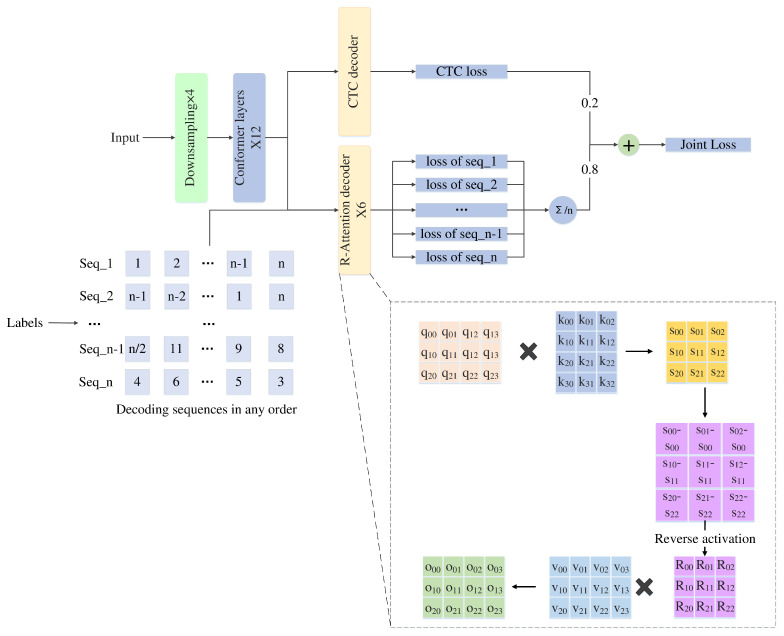
The overall workflow of the NLRD.

**Figure 2 sensors-24-03846-f002:**
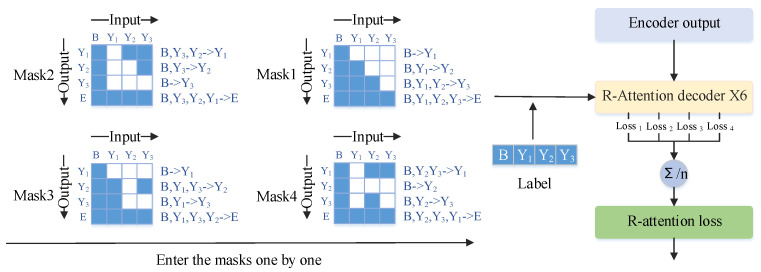
The specific computational process of NLT. We have illustrated only four masks, which sequentially participate in decoding, resulting in four loss values. The mean of these losses is then calculated as the loss for R-attention. Mask 1 represents the decoding mask for regular attention. To illustrate the working process of the masks, let us take Mask 4 as an example. When the input is B (the initial decoding token), we obtain the output Y2. Subsequently, by setting the input as B and Y2, we obtain the output of Y3. Adding Y3 to the input sequence, we obtain the output Y1. The operational principles of the other masks are similar.

**Figure 3 sensors-24-03846-f003:**
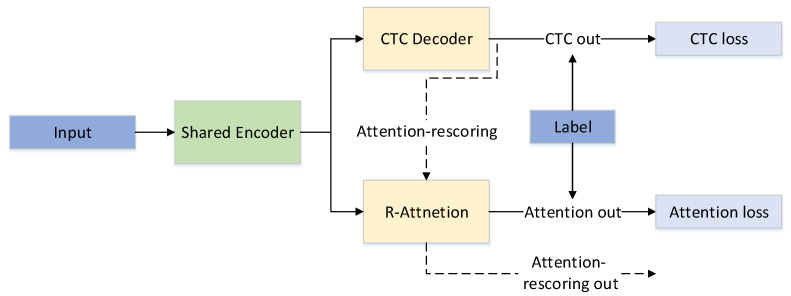
The NLRD-tiny model.

**Figure 4 sensors-24-03846-f004:**
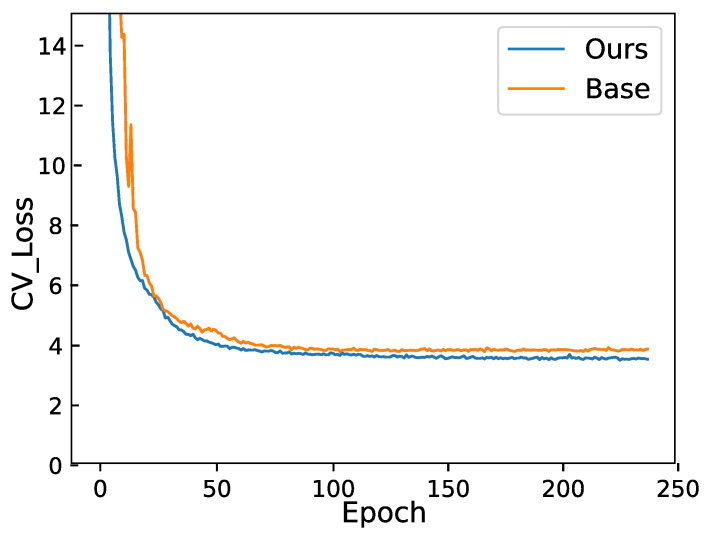
Validation loss on the Aishell1 dataset. The loss depicted in Figure is the validation loss on Aishell1. Our model referred to here is our NLRD model.

**Figure 5 sensors-24-03846-f005:**
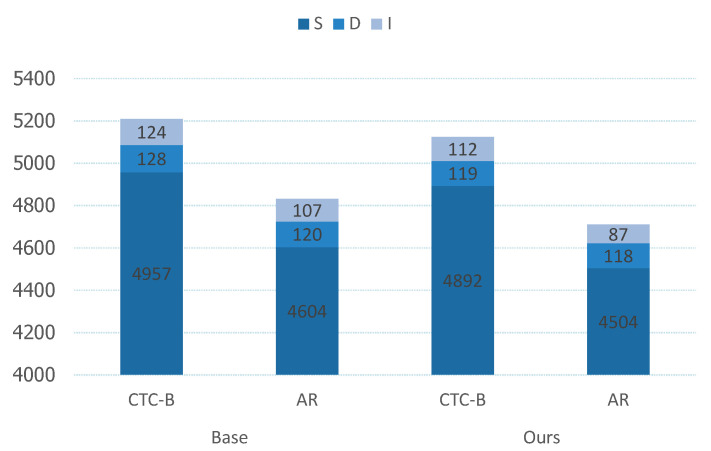
The distribution of recognition errors. The errors in Figure are all on Aishell1, and the error distribution refers to the count of errors made, where *S* represents Substitution Errors, *D* represents Deletion Errors, and *I* represents Insertion Errors.

**Figure 6 sensors-24-03846-f006:**
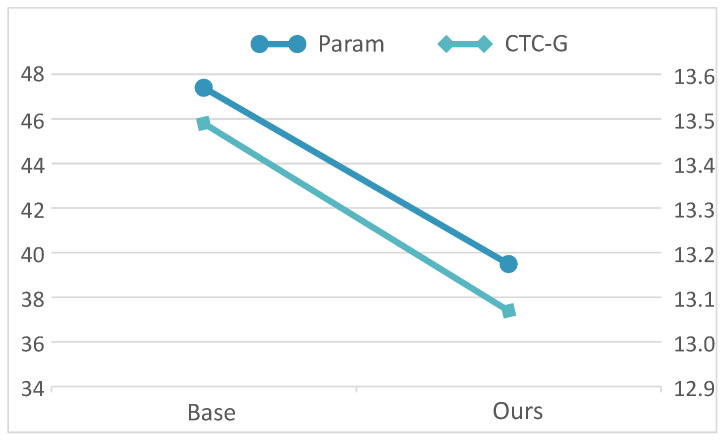
CER and param of NLRD-tiny. The data in Figure represents the performance on the Prime dataset, with separate plots illustrating the reduction in parameter count and the decrease in character error rate.

**Table 1 sensors-24-03846-t001:** Experimental parameter settings.

Parameter Name	Parameter Value for Chinese	Parameter Value for Ug
Num_Conformer	12	12
Num_Decoder	6	6
Dff	2048	2048
Epoch	240	120
Lr	0.002	0.0005
CTC_weigth	0.2	0.2
Num_Mel	80	80
Frame_Length	25	25
Frame_Shift	10	10
Spec_Aug	True	True
Accum_Grad	4	4

**Table 2 sensors-24-03846-t002:** Comparison with state-of-the-art speech recognition models, ↓ indicates that a lower error rate is better.

Model	CER on Aishell1 (↓)	CER on Prime (↓)	CER on ST (↓)	WER on Ug (↓)
Speech Transformer (2018) [9]	8.97	19.27	-	-
Transformer (Wenet) (2022) [10]	5.30	14.95	8.2	6.28
RNNT (2021) [40]	4.60	12.79	7.81	6.94
BAT (2023) [41]	5.28	15.56	8.56	6.57
NST (2022) [42]	4.85	12.97	7.63	-
Whisper (base) [43]	20.04	31.19	22.66	-
Whisper (large-v3) [43]	6.94	16.88	9.22	-
Paraformer (2022) [44]	5.20	15.02	7.89	-
Paraformer (U2) (2022) [10]	4.75	13.19	7.68	6.15
Efficient Conformer (2021) [18]	4.56	12.71	7.62	5.44
Conformer_Transformer (baseline) (2022) [10]	4.61	12.90	7.95	5.60
NLRD (Ours)	4.49	12.36	7.44	4.40
NLRD-tiny (Ours)	4.55	12.61	7.64	5.02

**Table 3 sensors-24-03846-t003:** Ablation studies on the NLT and R-attention modules are conducted on a model with 12 layers of the encoder and 6 layers of the decoder. En indicates the number of encoder layers, while De indicates the number of decoder layers, ↓ indicates that a lower error rate is better.

En	De	NLT	R-a	Attention-Rescoring (↓)		CTC-Greedy (↓)		CTC-Beam (↓)		Parameters (M)	
				Aishell1	Prime	Aishell1	Prime	Aishell1	Prime	Aishell1	Prime
12	6	*K* = 6	✔	4.49	12.36	4.89	12.97	4.89	12.97	46.2	47.4
12	6	*K* = 3	✔	4.56	12.48	4.93	13.17	4.93	13.17	46.2	47.4
12	6	✗	✔	4.59	12.86	4.91	13.46	4.91	13.45	46.2	47.4
12	6	*K* = 6	✗	4.52	12.51	4.90	13.07	4.90	13.07	46.2	47.4
12	6	✗	✗	4.61	12.90	4.98	13.49	4.98	13.49	46.2	47.4

**Table 4 sensors-24-03846-t004:** Ablation studies on the NLT and R-attention modules are conducted on a model with 6 layers of the encoder and 3 layers of the decoder. En indicates the number of encoder layers, while De indicates the number of decoder layers, ↓ indicates that a lower error rate is better.

En	De	NLT	R-a	Attention-Rescoring (↓)		CTC-Greedy (↓)		CTC-Beam (↓)		Parameters (M)	
				Aishell1	Prime	Aishell1	Prime	Aishell1	Prime	Aishell1	Prime
6	3	✔	✔	4.83	13.26	5.20	14.06	5.20	14.06	25.6	26.8
6	3	✔	✗	4.86	13.35	5.26	14.15	5.26	14.15	25.6	26.8
6	3	✗	✔	5.02	14.42	5.45	15.37	5.45	15.37	25.6	26.8
6	3	✗	✗	5.05	14.70	5.50	15.83	5.49	15.82	25.6	26.8

**Table 5 sensors-24-03846-t005:** The ablation experiments for NLRD-tiny. En indicates the number of encoder layers, while De indicates the number of decoder layers, ↓ indicates that a lower error rate is better.

En	De	NLT	R-a	Attention-Rescoring (↓)		CTC-Greedy (↓)		CTC-Beam (↓)		Parameters (M)	
				Aishell1	Prime	Aishell1	Prime	Aishell1	Prime	Aishell1	Prime
12	1	✔	✔	4.55	12.61	4.72	13.07	4.72	13.07	38.3	39.5
12	1	✔	✗	4.56	12.73	4.78	13.52	4.79	13.52	38.3	39.5
12	1	✗	✔	4.67	13.21	4.91	14.11	4.91	14.10	38.3	39.5
12	1	✗	✗	4.78	14.23	5.02	14.98	5.02	14.97	38.3	39.5

## Data Availability

The data that support the findings of this study are openly available in a public repository.

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
