# Peer review of "Nonlinear Regularization Decoding Method for Speech Recognition"

_sensors, 2024, doi:10.3390/s24123846_

Round 1

Reviewer 1 Report

Comments and Suggestions for Authors

The article proposes a methodology that integrates NLT and R-attention modules into an encoder-decoder architecture to improve speech recognition systems. While the article presents experimental results, it also suffers from several significant shortcomings.

Shortcomings:

1. The proposed NLRD method suffers from reduced training speed and increased GPU memory consumption due to multiple decoding steps, which hinders its scalability and efficiency, especially in resource-constrained environments.

2. The article fails to discuss the practical implications and real-world applications of the proposed methodology, neglecting to address issues such as scalability, deployment feasibility, and adaptability to diverse datasets (corpora), thereby limiting its relevance in practical speech recognition systems.

3. While the paper visualizes error distributions, it lacks a detailed analysis of the root causes of recognition errors, which limits the insights gained from the experiments and hinders the understanding of potential areas for improvement.

4. The comparative analysis with existing models lacks depth and fails to provide a nuanced understanding of the relative strengths and weaknesses of the proposed methodology compared to competing methods.

5. The authors consider only one modality for speech recognition, but it is known that speech recognition via audio modality may not work in noisy environments, for example. Therefore, it is reasonable to mention audiovisual speech recognition in the section with previous studies. But it is possible to write that at the moment audiovisual recognition is mainly done at the level of a limited vocabulary (see SOTA for audiovisual speech recognition on LRW: https://paperswithcode.com/sota/audio-visual-speech-recognition-on-lrw). And at the end add about AVCRFormer (Expert Systems with Applications - Q1), 2DCNN + BiLSTM + ResNet + MLF (Sensors - Q1), PBL (IEEE International Conference on Systems, Man and Cybernetics). This will show that the authors are up-to-date with currently known solutions and possible extensions of their work to multiple modalities. In addition, the links will be expanded to include highly rated journals.

Overall, the article deserves attention, but after eliminations and revisions.

Comments on the Quality of English Language

Moderate editing of English language required.

Author Response

Dear Expert

Thank you for your comprehensive review and valuable feedback on our submitted paper. Your professional insights have offered crucial guidance to our research, significantly influencing the enhancement of the paper's quality. We have diligently considered your suggestions and implemented corresponding revisions in the final version.

Once again, we appreciate your dedication and time spent reviewing our work. We eagerly anticipate receiving your guidance and suggestions in the future.

Best regards

Reviewer#1, Concern # 1: The proposed NLRD method suffers from reduced training speed and increased GPU memory consumption due to multiple decoding steps, which hinders its scalability and efficiency, especially in resource-constrained environments.

Author response: Dear expert, thank you for your careful guidance. Our NLT module consists of multiple decoding steps, but it is only used during training. In fact, we cannot anticipate subsequent decoding results during inference, so our model behaves similarly to a regular transformer decoder. The essence of our NLT module lies in decoding the same data in different orders multiple times to fully utilize the limited training data and enable the model to grasp contextual information from the text comprehensively. Therefore, under resource constraints, it offers advantages.

This viewpoint is well-supported as evidenced by the comparison between our approach NLRD and the baseline method Conformer_Transformer in Table 2 (page 10, Table 2, rows 11 and 12). Additionally, data from Table 3 demonstrates a decrease in error rates when using the NLT method (row 4) compared to the baseline method (row 5) (page 11, Table 3, rows 4 and 5).

Furthermore, we conducted statistics on the model's memory consumption and training time during the training process. Compared to the baseline, our model's training time increased by approximately 16%, and when K=6, memory usage increased by 9.2% (not explicitly stated in the paper). However, since NLT is not used during inference and testing, and the impact of the additional R-Attention in the model is negligible, it does not impose an additional burden during inference and testing.

Reviewer#1, Concern # 2: The article fails to discuss the practical implications and real-world applications of the proposed methodology, neglecting to address issues such as scalability, deployment feasibility, and adaptability to diverse datasets (corpora), thereby limiting its relevance in practical speech recognition systems.

Author response: Dear expert, thank you for your careful guidance. Regarding its practical implications and real-world applications, we have incorporated corresponding explanations in the introduction based on your professional advice (page 1, lines 24-29).

Addressing issues such as scalability, deployability, and adaptability to different datasets (corpora), our current research primarily focuses on projects related to Uyghur and Chinese speech recognition conducted in the laboratory. Subsequently, we plan to extend our research scope to include relevant studies in other languages.

Reviewer#1, Concern # 3: While the paper visualizes error distributions, it lacks a detailed analysis of the root causes of recognition errors, which limits the insights gained from the experiments and hinders the understanding of potential areas for improvement.

Author response: Dear expert, thank you for your guidance. Based on your professional advice, we have revamped the description of error analysis, elucidating the reasons for the improvements of our model over the baseline method. (page13-14, lines 507-521)

Reviewer#1, Concern # 4: The comparative analysis with existing models lacks depth and fails to provide a nuanced understanding of the relative strengths and weaknesses of the proposed methodology compared to competing methods.

Author response: Dear expert, thank you for your professional advice. Based on your professional advice, we have revised the comparative analysis in the Comparative Experiments section. (page 9-10, lines 345-389)

Reviewer#1, Concern # 5: The authors consider only one modality for speech recognition, but it is known that speech recognition via audio modality may not work in noisy environments, for example. Therefore, it is reasonable to mention audiovisual speech recognition in the section with previous studies. But it is possible to write that at the moment audiovisual recognition is mainly done at the level of a limited vocabulary (see SOTA for audiovisual speech recognition on LRW: https://paperswithcode.com/sota/audio-visual-speech-recognition-on-lrw). And at the end add about AVCRFormer (Expert Systems with Applications - Q1), 2DCNN + BiLSTM + ResNet + MLF (Sensors - Q1), PBL (IEEE International Conference on Systems, Man and Cybernetics). This will show that the authors are up-to-date with currently known solutions and possible extensions of their work to multiple modalities. In addition, the links will be expanded to include highly rated journals.

Author response: Dear Expert, thank you for your constructive suggestions. Based on your professional advice, we have added relevant discussion in the introduction section.(page1, lines 21-24, References[1-3])

Reviewer#1, Concern # 6: Moderate editing of English language required.

page 1, line 33

Original sentence: Moreover, it may require some prior knowledge, which is highly disadvantageous for the advancement of automatic speech recognition.

Modified sentence: Moreover, it may require some prior knowledge, which is highly disadvantageous for advancing automatic speech recognition.

page 2, lines 43-44

Original sentence: CTC decoding offers faster speed but overlooks contextual relationships, whereas attention decoding has a slower speed but higher decoding accuracy.

Modified sentence: CTC decoding offers faster speed but overlooks contextual relationships, whereas attention decoding has a slower speed but higher accuracy.

page 2, line 47

Original sentence: During the training process, the losses of both decoding methods are fused to guide the optimization of the encoder jointly.

Modified sentence: During the training process, the losses of both decoding methods are fused to jointly guide the optimization of the encoder.

page 2, lines 61-62

Original sentence: To address the aforementioned challenges, we propose the Nonlinear Regularization Decoding (NLRD) Method for Speech Recognition.

Modified sentence: We propose the Nonlinear Regularization Decoding (NLRD) Method for Speech Recognition to address the challenges above.

page 2, lines 67-69

Original sentence: To address the challenge of error accumulation during Attention-rescoring decoding, we introduce a Regularization Attention (R-attention) module.

Modified sentence: We introduce a Regularization Attention (R-attention) module to address the error accumulation challenge during Attention-rescoring decoding.

page 3, line 100

Original sentence: The Conformer model combined the advantages of Convolutional Neural Networks (CNNs) and Transformer models, using CNNs to extract detailed features and Transformer for global modeling, significantly improving the accuracy of speech recognition.

Modified sentence: The Conformer model combined the advantages of Convolutional Neural Networks (CNNs) and Transformer models, using CNNs to extract detailed features and Transformer for global modeling, significantly improving speech recognition accuracy.

page 3, line 102

Original sentence: Following that, Efficient Conformer and UCONV-CONFORMER made improvements to the complexity of Conformer to enhance the model's speed.

Modified sentence: Following that, Efficient Conformer and UCONV-CONFORMER improved the complexity of Conformer to enhance the model's speed.

page 3, line 107

Original sentence: These advancements have had a positive impact on the development of speech recognition technology.

Modified sentence: These advancements have positively impacted the development of speech recognition technology.

page 3, lines 113-114

Original sentence:  To address this issue, speech recognition models based on Long Short-Term Memory networks(LSTMs) decoding have been proposed.

Modified sentence: Speech recognition models based on Long Short-Term Memory networks(LSTMs) decoding have been proposed to address this issue.

page 3, line 122

Original sentence: while the cross-attention module is utilized to capture the relationship between the previous step's output and the encoder's output, enhancing the correspondence between sequences.

Modified sentence: In contrast, the cross-attention module captures the relationship between the previous step's output and the encoder's output, enhancing the correspondence between sequences.

page 3, line 138

Original sentence: In general, fusion models can often achieve better results.

Modified sentence: Fusion models can often achieve better results.

page 4, lines 142-143

Original sentence: In speech recognition, the decoder also employs fusion by sharing the encoder's input among different decoders.

Modified sentence: The decoder also employs fusion in speech recognition by sharing the encoder's input among different decoders.

page 4, lines 147-148

Original sentence: Due to the potential issue of gradient vanishing with LSTMs, a considerable amount of research tends to favor hybrid decoder models using CTC+Transformer.

Modified sentence: Due to the potential issue of gradient vanishing with LSTMs, much research favors hybrid decoder models using CTC+Transformer.

page 4, lines 152-153

Original sentence: However, hybrid decoders combine CTC loss and Transformer loss to maximize the model's generalization performance.

Modified sentence: However, hybrid decoders combine CTC and Transformer losses to maximize the model's generalization performance.

page 5, lines 168-172

Original sentence: The encoder utilizes the Conformer model, the decoder part consists of CTC decoding and Transformer decoding, while the Transformer decoding part of the decoder employs our designed NLRD, consisting mainly of NLT and R-attention modules. Additionally, to facilitate lightweight processing, we introduce a streamlined variant, NLRD-Tiny.

Modified sentence: The encoder utilizes the Conformer model. The decoder consists of CTC decoding and Transformer decoding. In contrast, the Transformer decoding part of the decoder employs our designed NLRD, which consists mainly of NLT and R-attention modules. Additionally, we introduce a streamlined variant, NLRD-Tiny, to facilitate lightweight processing.

page 6, lines 204-205

Original sentence: To enable the Transformer decoder to have the capability of decoding in any direction, we have designed a masking method.

Modified sentence: We have designed a masking method to enable the Transformer decoder to decode in any direction.

page 6, lines 214-215

Original sentence: In our practical experiments, a reversal operation is performed after obtaining K decoding sequences to strengthen the relationship between the preceding and succeeding sequences.

Modified sentence: Our practical experiments perform a reversal operation after obtaining K decoding sequences to strengthen the relationship between the preceding and succeeding sequences.

page 6, line 218

Original sentence: To illustrate with a specific example, if K is equal to 6, we effectively perform 12 decoding operations.

Modified sentence: To illustrate with a specific example, if K equal to 6, we effectively perform 12 decoding operations.

page 6, lines 220-222

Original sentence: This process helps optimize the model, enabling it to better understand and capture the correlation between preceding and succeeding sequences.

Modified sentence: This process helps optimize the model, enabling it to understand better and capture the correlation between preceding and succeeding sequences.

page 6, lines 224-225

Original sentence: We know that the Attention-rescoring decoder of Wenet uses the approach of taking the decoding results of beam search and passing them through the attention model for re-scoring.

Modified sentence: We know that Wenet's attention-rescoring decoder takes the decoding results of beam search and passes them through the attention model for re-scoring.

page 6, lines 226-228

Original sentence: There is a significant gap between these outputs and the actual output, which can have a substantial impact on the decoding of the attention model.

Modified sentence: There is a significant gap between these outputs and the actual output, which can substantially impact the decoding of the attention model.

page 6, lines 233-235

Original sentence: To address this challenge, we adopt a specific regularization method by controlling the attention scores within a certain range and placing more emphasis on the current input's impact on the results. The main difference between the R-attention model and the regular attention model lies in the regularization of attention scores.

Modified sentence: To address this challenge, we adopt a specific regularization method by controlling the attention scores within a certain range and emphasizing the current input's impact on the results.

The main difference between the R-attention and regular attention models lies in the regularization of attention scores.

page 6, lines 240-242

Original sentence: The purpose of this operation is to subtract the diagonal values of attention scores from their corresponding rows, resulting in an attention matrix with zeros along the diagonal.

Modified sentence: This operation aims to subtract the diagonal values of attention scores from their corresponding rows, resulting in an attention matrix with zeros along the diagonal.

page 7, lines 250-251

Original sentence: Our R-Attention will impose restrictions on this information to prevent errors from accumulating in the subsequent output.

Modified sentence: Our R-Attention will restrict this information to prevent errors from accumulating in the subsequent output.

page 7, lines 259-260

Original sentence: During the training process, the loss calculation formula we used is given by equation (4).

Modified sentence: The loss calculation formula we used is given by equation (4) during the training process.

page 7, lines 263-264

Original sentence: This is because the combined loss effectively provides complementary information for two decoding approaches.

Modified sentence: The combined loss effectively provides complementary information for two decoding approaches.

page 7, lines 271-273

Original sentence: It is worth noting that during the inference process, we only use forward decoding, similar to regular attention models, and it does not increase the inference time.

Modified sentence: It is worth noting that we only use forward decoding during the inference process, similar to regular attention models, and it does not increase the inference time.

page 7, lines 276-277

Original sentence: The Shared Encoder is a shared encoder utilized for both CTC decoding and attention decoding.

Modified sentence: The Shared Encoder is a shared encoder utilized for CTC and attention decoding.

page 8, lines 291-292

Original sentence: All utterances have been transcribed and meticulously checked by humans to ensure transcription accuracy.

Modified sentence: Humans have transcribed and meticulously checked all utterances to ensure transcription accuracy.

page 8, lines 299-300

Original sentence: Excluding the test set and validation set, we incorporate all remaining data into the training set.

Modified sentence: Excluding the test and validation datasets, we incorporate all remaining data into the training dataset.

page 8, line 304

Original sentence: Therefore, for Chinese datasets, we use single Chinese characters as the tokenization standard and include "black" as well as start/end tokens in the vocabulary.

Modified sentence: Therefore, for Chinese datasets, we use single Chinese characters as the tokenization standard and include "black" and start/end tokens in the vocabulary.

page 8, lines 306-307

Original sentence: For the UG dataset, we employ the publicly available Byte Pair Encoding (BPE) method for vocabulary generation.  

Modified sentence: We employ the publicly available Byte Pair Encoding (BPE) method in the UG dataset for vocabulary generation.

page 8, line 312-313

Original sentence: The differences in vocabulary among the four datasets result in slightly different model sizes, albeit the variances are minor.  

Modified sentence: The differences in vocabulary among the four datasets result in slightly different model sizes, albeit minor variances.

page 8, lines 327-328

Original sentence: In our experiment, due to the difference in speech lengths between the four datasets, the allowable batch size varies.  

Modified sentence: In our experiment, the allowable batch size varies due to the difference in speech lengths between the four datasets.

page 9, lines 329-330

Original sentence: The settings for other hyperparameters are shown in Table 1.  

Modified sentence: Table 1 shows the settings for other hyperparameters.

page 10, lines 390-391

Original sentence: To demonstrate the capability of our model in extracting contextual information and the implementation of lightweight models, we have introduced a tiny model.  

Modified sentence: To demonstrate the capability of our model in extracting contextual information and implementing lightweight models, we have introduced a tiny model.

page 10, lines 394-395

Original sentence: Compared to the baseline, our tiny model achieved reductions in error rates of 0.06%, 0.29%, 0.31%, and 0.58%, respectively.  

Modified sentence: Compared to the baseline, our tiny model reduced error rates of 0.06%, 0.29%, 0.31%, and 0.58%, respectively.

page 11, lines 407-408

Original sentence: In our NLT module, the number of decoding steps, denoted as K, has a significant impact on the experiments.  

Modified sentence: In our NLT module, the number of decoding steps, denoted as K, significantly impacts the experiments.

page 11, line 422

Original sentence: From Table 3, we can observe that both modules play a positive role in the overall system.  

Modified sentence: Table 3 shows that both modules play a positive role in the overall system.

page 11, lines 424-426

Original sentence: This is because, with an increase in the number of decoding steps, the system can better capture the correlation between texts, extracting more semantic information.  

Modified sentence: This is because, with increased decoding steps, the system can better capture the correlation between texts, extracting more semantic information.

page 11, lines 430-431

Original sentence: In our experiment, if we set K=9, an error exceeding the memory capacity will occur.

Modified sentence: In our experiment, an error exceeding the memory capacity will occur if we set K=9.

page 12, lines 458-460

Original sentence: This not only reduces parameter count but also allows the R-attention to compensate for the loss of contextual information during CTC decoding, thereby improving the effectiveness of the CTC decoding process.  

Modified sentence: This reduces parameter count and allows the R-attention to compensate for the loss of contextual information during CTC decoding, thereby improving the effectiveness of the CTC decoding process.

Reviewer 2 Report

Comments and Suggestions for Authors

The paper introduces a novel Nonlinear Regularization Decoding (NLRD) method for speech recognition, which addresses some of the key challenges in the field, such as error accumulation and complexity with small datasets. The methodology is well-defined, with clear explanations of the NLT module and R-attention module.  The paper provides a detailed description of the computational process of the NLT module and the R-attention mechanism, which helps in understanding the model's working. The comparison with existing models and the ablation studies further validate the contribution of the NLT and R-attention modules. While the paper is thorough, it might benefit from a broader discussion on how the NLRD method compares to state-of-the-art methods in terms of computational efficiency and scalability.

Author Response

Dear Expert

Thank you for your comprehensive review and valuable feedback on our submitted paper. Your professional insights have offered crucial guidance to our research, significantly influencing the enhancement of the paper's quality. We have diligently considered your suggestions and implemented corresponding revisions in the final version.

Once again, we appreciate your dedication and time spent reviewing our work. We eagerly anticipate receiving your guidance and suggestions in the future.

Best regards

Reviewer#, Concern #:  While the paper is thorough, it might benefit from a broader discussion on how the NLRD method compares to state-of-the-art methods in terms of computational efficiency and scalability.

Dear expert, thank you for your careful guidance. Our NLT module consists of multiple decoding steps, but it is only used during training. In fact, we cannot anticipate subsequent decoding results during inference, so our model behaves similarly to a regular transformer decoder. The essence of our NLT module lies in decoding the same data in different orders multiple times to fully utilize the limited training data and enable the model to grasp contextual information from the text comprehensively. Therefore, under resource constraints, it offers advantages.
This viewpoint is well-supported, as evidenced by comparing our approach NLRD and the baseline method Conformer_Transformer in Table 2 (page 10, Table 2, rows 11 and 12). Additionally, data from Table 3 demonstrates a decrease in error rates when using the NLT method (row 4) compared to the baseline method (row 5) (page 11, Table 3, rows 4 and 5).
Furthermore, we conducted statistics on the model's memory consumption and training time during the training process. Compared to the baseline, our model's training time increased by approximately 16%, and when K=6, memory usage increased by 9.2% (not explicitly stated in the paper). However, since NLT is not used during inference and testing, and the impact of the additional R-Attention in the model is negligible, it does not impose an extra burden during inference and testing.

Based on your professional advice, we have revised the comparative analysis in the Comparative Experiments section. (page 9-10, lines 345-389)

Round 2

Reviewer 1 Report

Comments and Suggestions for Authors

The authors completed the article. All shortcomings have been successfully corrected and questions have been answered. In this form, the article looks complete and can be recommended for publication.

Comments on the Quality of English Language

Minor editing of English language required.